# The Governance of Traffic Noise Impacting Pedestrian Amenities in Melbourne Australia: A Critical Policy Review

**DOI:** 10.3390/ijerph21081080

**Published:** 2024-08-16

**Authors:** David O’Reilly, Marcus White, Nano Langenheim, Pantea Alambeigi

**Affiliations:** 1Hawthorn Campus, Department of Interior Architecture & Industrial Design, Centre for Design Innovation; Swinburne University of Technology, John St, Hawthorn, VIC 3122, Australia; marcuswhite@swin.edu.au (M.W.); palambeigi@swin.edu.au (P.A.); 2Parkville Campus, Melbourne School of Design, University of Melbourne, Masson Rd, Parkville, VIC 3010, Australia; nano.langenheim@unimelb.edu.au

**Keywords:** traffic noise pollution, pedestrian amenities, health impacts, noise management, governance frameworks, policy interventions, qualitative noise data, walkability, environmental acoustics

## Abstract

By identifying a unified aim of Federal, State, and Local government authorities to deliver healthier, more liveable urban spaces and enable walkable neighbourhoods in Melbourne, Australia, questions emerge regarding noise data collection methods and the policies that aim to protect pedestrian areas from potential increases in urban traffic noise. It highlights a missed opportunity to develop strategies that provide explicit guidance for designing more compact urban forms without diminishing pedestrian amenities. This study investigates the governance of traffic-induced noise pollution and its impact on pedestrian amenities in Melbourne, Australia. It aims to identify the government bodies best positioned to protect pedestrians from noise pollution and evaluate the strategic justification for reducing traffic noise to enhance urban walkability. This research employs a semi-systematic policy selection method and a hybrid critique and review method to evaluate the multidisciplinary governance frameworks engaged in the management and mitigation of traffic noise in Melbourne. Key findings reveal that while traffic noise poses significant health risks, current policies overlook its impact on pedestrian amenities in urban areas. This study emphasises the benefits of qualitative and subjective noise data collection to inform policy-makers of the pedestrian aural experience and impacts. Discussion points include noise management strategies and the value of implementing metropolitan-scale noise-mapping to illustrate the impact of noise rather than quantities of sound. The conclusions demonstrate that there is strategic justification for managing traffic-induced noise pollution to protect pedestrian areas within international, federal, and state government policies and implicit rationale at a local level.

## 1. Introduction

Globally, urban noise pollution is recognised as a significant public health problem [1]. Concurrently, walking is also recognised as a preventative health approach that counters these very same adverse effects [2] (pp. 515–517). The European Union’s ‘Frontiers-Noise, Blazes and Mismatches’ document claims that long-term exposure to environmental noise causes 12,000 premature deaths and 48,000 new cases of ischemic heart disease annually in Europe, with over 22 million people suffering from chronic noise annoyance [3] (p. 9). The World Health Organisation [WHO] also considers traffic-induced noise pollution to be a public health issue, contributing to cardiovascular disease, diabetes, obesity, cancer, depression, and stroke, often linked to sleep disturbance [1,4]. Coincidentally, walking is recognised as a preventative measure for some these conditions [5,6].

Unfortunately, in most urban environments, pedestrians are highly exposed to traffic noise with minimal opportunity to attenuate their exposure. Noise pollution has historical roots, with the first noise abatement ordinance occurring in the first century BC [7] (p. 2). Since then, many cities have been redesigned to at least accommodate, if not prioritise vehicle access and movement, often at the expense of pedestrian amenities [8], human health [5] (p. 3), [6] (p. 3), [9,10] and the environment [7] (p. 6), [8] (p. 1294), [11,12]. This shift in design priorities has exacerbated noise pollution and is now considered a significant global problem [3] with serious health and social implications [13] (p.783) that inadvertently diminish the quality of a pedestrian experience, with potential to alter route preferences [14] (p. 356). Ren et al. [15] believe that traffic negatively impacts the acoustic environment, which in turn, has health implications, including psychological discomfort. Their belief is supported by other studies that demonstrate that environmental noise is perceived as an environmental stressor that creates a nuisance in urban environments and impacts an individual’s walking preferences [14,16]. Mousavimasouleh et al. [17] (p. 8) observed that that air pollution and noise pollution were the two most important criteria for people when choosing a trip mode. They claim that from a pedestrian perspective, the issues of air and noise pollution rank higher than velocity, accessibility, and comfort in determining the quality of a walking experience and conclude that transport experts and decision-makers are obliged to increase the attractiveness of active transport modes to protect an individual’s health and the environment so that traffic noise pollution is not exacerbated by people paradoxically opting to drive in an effort to avoid air and noise pollution.

The Charter for Human Rights and Responsibilities Act 2006, Freedom of Movement, asserts that every person lawfully within Victoria has the right to move freely within the state [18] (p. 13). However, ‘move freely’ does not imply without cost, which economically, can only be achieved by walking. Walking is the most affordable, sustainable, and healthy mode of travel, requiring minimal infrastructure [19] and, by protecting amenities from the impacts of noise, might increase the likelihood that more people will choose to walk rather than drive on short journeys [17]. Pedestrians have minimal protection from adjacent car traffic, especially when infrastructure is built close to the carriageway where exposure to air and noise pollution is most likely to diminish their experience [8]. Consequently, pedestrians may seek alternative routes or take up less sustainable modes of transport [6,19,20,21]. Recent studies demonstrate that noise is a significant urban irritant for the residents of Melbourne [22], with traffic noise identified as the primary source of annoyance [4] (p. 2). Moreover, noise pollution is considered the second most harmful environmental factor to public health, surpassed only by air pollution [23], while the populations most affected by traffic noise are pedestrians living active lifestyles or commuting in the most sustainable way [24].

This study focuses on Melbourne, in the state of Victoria, Australia. Melbourne’s urban context was chosen for this review due to its high liveability [25] and the State government’s agenda to restructure neighbourhood activity centres into walkable, 20-minute neighbourhoods [26] that could place pedestrian and residential amenities under threat by excessive traffic noise due to predicted 65% increases in freight traffic across Melbourne’s transport network by 2050 [27].

As the world’s population continues to grow and more people are living in urban areas [20,28], the method used in this study may prove useful in other parts of the world. To understand specific conditions in other regions, the hybrid critique and policy review method can be modified to fit other contexts that would reveal different, but equally useful results. The benefit of this approach is that it considers noise pollution beyond the emissions from the source and provides understanding of how it impacts a population such as pedestrians. Today, Melbourne is ranked as one of the world’s most liveable cities, attributed to its culture, environment, education, and infrastructure [25], and, as with other cities wanting to protect liveability standards, the quality of the environment plays a crucial role in making walking a viable, healthy, and sustainable alternative to driving [14], [29,30,31]. Kasagici and Nuray Ates [32] (p. 1) claim that the quality of pedestrian infrastructure is vital to encourage more people to live active lifestyles, but without improved amenities, the enjoyment of active transportation is diminished, eroding the potential benefits of the State government’s plans to restructure neighbourhood activity centres into healthy, high amenity, walkable, 20-minute neighbourhoods [26,33,34].

The Australian Bureau of Statistics medium series predictions for Victoria’s population is projected to grow from 6.9 million in 2024 to 10 million by 2055 [35]. To facilitate this projected growth, State and Local governments have begun trials restructuring existing Neighbourhood Activity Centres [NACs] across metropolitan Melbourne into walkable, 20-minute neighbourhoods [36] (p. 98). For this strategy to succeed sustainably, it requires systems thinking of and monitoring pedestrian amenities while managing traffic, higher density housing, employment, and community infrastructure [37,38], to ensure that people are not just spatially capable of walking short journeys but want to. Compact city structures encourage walking over driving for short trips, reducing traffic and emissions, and enhancing community health and resilience [39]. As traffic noise has health implications and there are plans to restructure Melbourne’s activity centres into walkable, 20-minute neighbourhoods, there is a need to understand the effectiveness of Melbourne’s governance framework for protecting pedestrians from potential increases in traffic-induced noise pollution.

The governance of Victoria’s public sector decision-making is outlined in The Local government Act 2020 [40]. It requires a holistic utilitarian approach that is multidisciplinary in nature and requires transparency and community engagement as key components of the governance framework. This is to ensure decision-making is performed fairly, is accountable to the community, and is based on merit. To enable good governance, understanding the impacts on a community is critical. Decisions need to be informed by research, and government intervention must also be in the interest of improved community outcomes. On the issue of noise, empirical data in the form of metropolitan-scale noise maps can present the impacts of noise subjectively, which can be useful for public sector decision-making by providing greater understanding of pollution when restructuring the road hierarchy and land uses at a metropolitan scale.

With this context in mind, the aim of this paper is to evaluate the existing governance of noise by critiquing the effectiveness of existing policies and identifying possible insufficiencies, such as the protection of pedestrian amenities, protocols to create metropolitan-scale noise mapping, and subjective data collection methods that are already used to measure noise impacts in public spaces in other parts of the world [41,42,43,44].

## 2. Method

The hybrid policy review and critique method used for this review was chosen to ensure the governance of noise is reviewed from a multi-disciplinary perspective rather than noise policies alone. This allows us to consider not just the emissions of noise, but its impacts by including community-outcome-focused policies that inform decision-making within the governance framework and that stretch beyond the field of acoustics. The hybrid approach was chosen to remove subjectivity and provide a more detailed understanding of the document content that extends beyond the critique alone. In order to address the aim outlined in the introduction, and with respect to the broader scope applied to public sector governance in Melbourne, the policies and documents that were considered part of the governance framework include advocacy and research, as well as policies related to noise, the environment, health, and transport at the Local, State, and Federal levels.

This method and inquiry-driven approach created two key components. The first component is the policy review, which includes an evaluation and content analysis. This component evaluates the focus of the screened documents that are used to guide decision-making and policy development. The purpose is to evaluate their effectiveness at protecting pedestrian amenities in urban areas of Melbourne and to identify if they have inferred meaning such as protecting a community in a broader sense than noise impacts alone. The second component is a policy critique that examines the aims and objectives of the policies and whether or not they seek improved outcomes in three sub sections: Environment, Health, and Transport. The results of this component identify if the policies are explicitly focused on noise, if they provide strategic justification to protect pedestrians from the impacts of noise, or if they recommend noise mapping, or data collection methods that can be used to inform decision-making.

The multidisciplinarity of stakeholders and the combined interest in mitigating noise and improving walk quality required exploration into the policies and reports of numerous authorities that govern noise, public space, access and movement, and development in Melbourne and more broadly, Australia. Furthermore, international protocols and advocacy documents were identified within the governance structure from citations within the eligible policies. This method introduced policies on the issue of noise pollution beyond the field of acoustics by considering the impacts on active transportation, community health, the environment and, in particular, from the perspective of the pedestrian as receiver, rather than focusing on the source of noise or residential amenities.

As governance in Australia’s public administration involves a network of participants [45] (p. 17), documents have been sought from various tiers of government and the authorities appointed to manage relevant components, such as Victoria’s Environmental Protection Agency [EPA]; VicRoads, the authority responsible for managing Victoria’s major roads; Victoria’s Department of Transport and planning; as well as the State and Federal Departments of Health. Furthermore, good governance in the public sector extends to non-government and community sectors for equity and inclusiveness [46]. To ensure governance is considered in the broader authorising environment, advocacy documents from non-governing organisations are included to ensure their participation, independent research, and advocacy is considered for its influence on and contribution to decision-making and policy development. The non-governing organisations considered most relevant include The World Health Organisation [WHO], which globally conducts research and advocacy for managing health-related issues, including noise [41]; The Environmental Health Standing Committee [enHealth], which is appointed to provide advice to the Federal government in Australia on environmental factors affecting health, including noise [47]; The Heart Foundation Australia, a national health promotion charity and advocacy group that promotes walking as a preventative health initiative [48]; The Municipal Association of Victoria [MAV], appointed to deliver high-quality governance across the local government sector [49]; and Victoria Walks, an independent, evidence-based health promotion charity organisation that provides research and advocacy to state and local government authorities to promote greater participation in walking and health [50].

### 2.1. Method for Document Screening and Selection

The policy content was selected using a semi-systematic approach that combines a structured search strategy using the ‘Preferred Reporting Items for Systematic reviews and Meta-Analyses–Search’ [PRISMA–S] to screen policies from government libraries, Local government Council plans, and advocacy and research that were used to inform policy directions. The PRISMA–S method is an extension of the PRISMA method, which provides reporting of systematic reviews and meta-analysis with the PRISMA–S check-list, provided in the Appendix A. The PRISMA–S method is a more flexible alternative method to the PRISMA checklist and reporting method. The PRISMA–S method provides flexibility and broad inclusion alongside the benefits of rigour and credibility [51,52,53]. The PRISMA–S application in this review was preferred to the standard PRISMA reporting method due to the multidisciplinarity of the study extending into numerous fields and across a range of governing bodies and authorities appointed by three tiers of governments and the need to access grey literature and acquisition of Council plans using a spatial screening criterion. Relevant themes for the document search were identified using the PICo qualitative framework, a complimentary process used to define the scope of the PRISMA–S search criteria by pre-defining a population, interest, and context. The PICo qualitative framework has pedestrians as the ‘population’; authorities that govern transport, health, environment, and noise as the ‘interest’; and the urban areas of Melbourne Australia as the ‘context’ applied to.

Document searches were conducted from Federal, State, and Local government document libraries as well as Transport, planning, Environment, and Health Department policy directories. The search terms included the following: environment/environmental noise, transport/transportation noise, health noise AND at least one of the following words: policy, plan, provision, regulation, instrument, act, guidelines, standards, surveys, programmes, strategies, frameworks, report, and protocol. The identified policies were acquired and screened through active reading for relevance, duplication, and currency. The document search was limited to draft and current policies related to traffic, noise, vehicle standards, planning, health, and environments that apply to Melbourne Australia. In addition, noise policies and reports from the World Health Organisation, European Union, International Standards, and local advocacy groups were included in the results. These were identified through references and citations identified within Australian policies and guidelines.

In Victoria, Local government Council plans are the over-arching decision-making framework that is underpinned by a community vision and strategic directions, which provides a transparent reference for decision-making related to expenditure, operational matters, capital works, and public realm improvements for the 4-year term of office. With 79 local government areas [LGAs] in Victoria, Australia, and more than 30 in the metropolitan urban growth boundary, a separate screening method was applied for the purpose of identifying the LGAs most likely to present an urban character. As outlined in the Local government Act 2020 [40], each municipality must draft their own Council plan, so document selection would not be possible from a central document library. According to the State government, neighbourhood activity centres have potential to be restructured as high-density, walkable, 20-minute neighbourhoods and were subsequently used to screen the document pool with respect to the research aim. A spatial criterion was used to identify local government areas that present urban characteristics and neighbourhood activity centres within 20 km of Melbourne’s General Post Office [GPO]. Council plans were acquired from their individual websites and screened to highlight vision statements and strategic directions that aim to improve environment, transport, and health outcomes where noise pollution could be considered a problem, or its measurement considered a key performance indicator. Excluded policies were either superseded, duplicated, deemed unrelated to noise pollution, pedestrian amenities, or not specific to Melbourne. To contain the scope of the study, provisions focused on aircraft, windfarms, live music, animals, land uses, as well as Council plans beyond a 20 km radius of Melbourne’s General Post Office were also excluded from the study, as they were not considered to impact pedestrian’s subjectivity or willingness to walk compared to traffic-induced noise pollution.

### 2.2. Method for Policy Review

The policy review involved active reading and word searches of the screened documents to assess the aims and purpose and to understand the elements and principles used to guide decision-making and policy development. The purpose of this was to evaluate their effectiveness at protecting pedestrian amenities in urban areas of Melbourne as a guide to decision-making. The documents’ contents were evaluated for explicit directions on noise, implicit directions to protect pedestrian amenities, or strategic directions aimed at improving the health, environment, or transport systems using the nine themes illustrated in Table 1.

This method allowed the review to consider governance more broadly by considering policies that seek to contain outputs of noise or deliver outcomes for the community. For policies that do not provide explicit content relating to noise or pedestrian impacts, implicit strategic justification was considered by examining the strategies related to amenities in general, transportation, health, and the environment that are all impacted by traffic noise. 

Following the initial policy evaluation method, the Local government Council plans were also reviewed using a relational qualitative content analysis approach [54] due to the complete lack of references to noise. This method is typically used for systematically analysing written, verbal, or visual documentation to identify key themes from a variety of sources or to gain understanding or meaning of text when it is not overtly apparent [55]. The relational qualitative content analysis approach also maintains more statistical rigour than other qualitative methods, including grounded theory, narrative analysis, or participatory action research [56].

### 2.3. Method for Policy Critique

Following the document screening and evaluation, the policy critique involved active reading and a written summary of the policies content. The purpose of this component is to consider the effectiveness, focus, or possible insufficiencies of the document, whether noise is explicitly referred to in the document or not. The included noise policies were also evaluated for spatial reference to pedestrian areas to identify if guidelines exist for noise modelling in footpaths, paved areas, or public open spaces. Furthermore, the review considers the noise-measurement and -monitoring protocols to identify if they recommend using objective noise metrics, such as decibels, which measure the source of noise; subjective indices designed to measure the impact on the receiver, such as the soundscape approach, commonly used to assess environmental noise [3,43,57]; or psychoacoustics, which offer noise annoyance metrics, commonly used to measure the tonal qualities of sound and how they subjectively impact the receiver [58,59].

## 3. Results

### 3.1. Results of Document Screening and Selection

The PICo qualitative framework identified the ‘population’ to be pedestrians; the ‘interest’ group as authorities that govern transport, health, environment, and noise; meanwhile, the ‘context’ applied to the urban areas of Melbourne Australia. In response to the PICo framework and the research aim outlined in the Introduction, mixed methods for document identification were used. Firstly, identification of local government areas identified 31 LGAs within Metropolitan Melbourne and 23 LGAs within 20 km of Melbourne’s GPO. The 23 Council plans were selected using a 20 km GIS buffer as pictured in Figure 1. Each identified LGA’s Council plan was individually acquired from the Council’s website.

Secondly, searches of the Federal and State government websites and their authorities initially identified 1391 potential policy documents. These were screened and selected as either a policy, plan, provision, regulation, instrument, act, guidelines, standards, surveys, programmes, strategies, frameworks, report, or protocol. Superseded documents; duplicate documents; those unrelated to noise pollution, traffic, or pedestrian amenities; those not used in Melbourne’s urban context; and Local government Council plans beyond 20 km of GPO were removed. A final list of 53 documents were included in the study. These documents were mostly from Australia, with 3 from abroad and another 3 from non-government organisations advocating for improved health and active transport. The rest were published by Australia’s Federal, State, and Local government organisations and their authorities. The list of 53 documents included in the study is provided with the evaluation in the Appendix A, while the process that produced the result is illustrated in the document-selection flow diagram, Figure 2.

### 3.2. Results of Policy Review

A relational qualitative content analysis approach was applied to review two main elements within each Council Plan: firstly, a vision statement drafted in consultation with their local community, and secondly, a set of strategic directions outlining the community outcomes they claim will realise the aspirations outlined in the vision statement. The results of the relational qualitative content analysis demonstrated that all of the 23 Council plans had numerous strategic directions that seek improved environment, health, and transport outcomes, although none of them presented or even referenced the issue of traffic noise.

Despite the Council plans not referencing traffic noise, each strategic direction was assessed and considered for if noise metrics could be applied as a performance indicator, or if the strategy could be realised, in part, by a reduction in traffic noise. The results show there were significant correlations between the Council plan’s aims to provide healthy, safe, and more liveable urban environments, and it is possible that a reduction in traffic noise could achieve the desired community outcomes in part. On average, each LGA had 10.5 strategic directions seeking improved environmental outcomes, 8 seeking improved health outcomes, and 6.7 supporting active transport or more efficient transportation outcomes. The average total of strategic directions referencing the three themes implied that each LGA had provided 25 strategic directions within their Council Plan with potential strategic justification to consider the impacts of traffic noise implicitly. The complete content analysis of the Council plans can be found in the Appendix A.

Table 2 illustrates a sample extract and the results of the content analysis, as interpreted through the process. After the initial search for content, it was noted that in the absence of noise references, all LGAs had numerous strategies aimed to improve environment, health, and transport outcomes. These were tabled as themes, and the contents were assessed through active reading. Inferences that may provide strategic justification for reduced noise pollution or could be informed by noise metrics as a performance indicator were highlighted according to the three themes, with inferences awarded a point and totals presented for each Local government Council plan. The results demonstrate that noise metrics could be used to inform the performance of policies or even the Council plan itself. It also demonstrates that there is strategic justification to reduce traffic-induced noise pollution with the aim of improving walkability across multiple local government areas in Melbourne.

Extending the content analysis to consider the focus of the 53 documents, including the Council plans, policies, advocacy, and research, a summary of the document evaluation is displayed in Table 3. The summary table of the document evaluation shows that the majority of documents did not present advocacy for a reduction in traffic noise, a focus on noise, or a focus on pedestrian amenities. They did, however, present explicit focus on community outcomes with a strong focus on health, the environment and transport/traffic as well. As a reduction in traffic noise can improve community, health, environment, and transportation outcomes, it can be concluded that monitoring and mapping of traffic noise impacts could be used as a key performance indicator for numerous plans and policies by providing supporting data that could be used as a key performance indicator to inform a public sector decision-making framework.

All the documents reviewed provided strategies or recommendations that aim to protect or enhance amenities in the interest of community well-being. Most focus on improving or maintaining residential and public space amenities, while 74% identify the need to protect or enhance pedestrian amenities. The results demonstrate that, even though the Council plans do not refer to noise pollution explicitly, they have a strong focus on community outcomes including health, the environment, and transport that could be negatively affected by noise impacting pedestrians.

### 3.3. Results of Policy Critique

#### 3.3.1. International Benchmarks

The association of transport agencies, Austroads, represents all levels of government transport authorities in Australia and New Zealand. They published ‘An approach to the Validation of Road Traffic Noise’ [60] that suggests the UK Calculation of Road Traffic Noise [CRTN] model for measuring traffic noise is commonly used in Australia for its simplicity. However, the CRTN is primarily focused on the outputs of noise using objective metrics to quantify emissions from the source. Nor does it offer guidelines for the creation of metropolitan-scale strategic noise-mapping methods or contemporary modelling approaches for understanding noise impacts in public spaces or pedestrian areas [61,62,63,64,65,66]. The lack of accessible noise data at the metropolitan scale makes it difficult to assess the quality of environmental noise or knowing where the problem exists and where its impact needs to be addressed. This sees noise audits being conducted only in response to a noise complaint, noise abatement notice, or on a site-by-site basis that has no strategic value for public governance or planning.

The European Union adopts a centralised approach to traffic noise that provides common noise assessment and noise-mapping methods across member states as described in the European Union Common Noise Assessment Methods [CNOSSOS-EU] framework [42] and the JRC Reference Report [67,68,69]. The CNOSSOS-EU framework is focused on objective metrics for standardised noise mapping. In contrast, the WHO provides research and guidelines for monitoring noise that is designed to capture its impact on the environment and health using subjective indices rather than decibels for application in public spaces [3,41]. The WHO advocates for a soundscape approach to noise monitoring, which provides understanding of the environmental impact of traffic noise in public spaces using descriptive classifications that are easier to understand than the physics of sound. This method is both useful to understand the impacts on pedestrians, and easier for decision-makers to assess by focusing on contextual relationships between pedestrians, transportation, and land uses that are not automatically apparent by quantifying emissions [3,4,43].

#### 3.3.2. Are Pedestrians Considered in Melbourne’s Noise Governance Structure?

The State government’s plan Melbourne strategy provides strategic justification for high-amenity, walkable, compact neighbourhoods [36]. Overall, the agenda focuses on liveability with the goal of fostering healthy, sustainable communities through well-designed walkable neighbourhoods by connecting a mix of land uses, housing types, and improved access to public transport and community infrastructure [26]. plan Melbourne 2017–2050 is the overarching long-term metropolitan-scale strategy that emphasises [in objective 6] the need to improve air quality and reduce excessive noise impacts [36] (pp. 10–11). The summary document highlights that more people in mixed-use areas could be exposed to pollution and recommends technical guidance be considered early in development processes so that any future population and housing growth plans deliver high-quality public spaces with particular attention to pedestrian amenities. Enhancing pedestrian amenities in walkable neighbourhoods involves integrating land use and transportation, as outlined in the Transport Integration Act 2010 [70] and VicRoads Movement and Place Framework [71]. For an effective response to traffic noise, plan Melbourne highlights the need for noise mitigation strategies as a structure-planning and urban design issue for all responsible and referral authorities to consider when proposing plans and projects that impact pedestrian spaces and overall community health.

The Federal government recently released a new Draft National Urban Policy for community consultation [72] that supports the aspirations outlined in plan Melbourne. The draft provides numerous objectives with no direct reference to noise. However, it does highlight that Australia’s cities should be safe to travel in and implies that a well-built city promotes active, independent living. Alternatively, it claims that a lack of safe and accessible walking and riding paths contributes to adverse health outcomes and social isolation and highlights that walking and cycling infrastructure can counter these problems and enhance liveability and productivity while reducing congestion and vehicle emissions. The policy concludes that increased active travel within communities requires behavioural shifts to achieve the objectives [72] (pp. 17–37) and one way to do this is by improving pedestrian amenities [8].

Another policy of equal significance that is not focused on pedestrian amenities is the State government of Victoria’s Public Health and Wellbeing Act 2008 [73]. This Act is the strategic tool that activates a Local government duty to remedy environmental noise nuisances that have impacts on community health. The responsible authority positioned under Part 6 of the Public Health and Wellbeing Act 2008 [73] (pp. 69–70) states that Local governments must consider noise as ‘nuisances which are, or are liable to be, dangerous to health or offensive’, and that ‘Council has a duty to remedy as far as is reasonably possible all nuisances existing in its municipal district’. Under the Act, Councils must investigate any notice of nuisance, and if the person that filed a complaint was not satisfied with the response, it can be addressed by the municipal Magistrates court for review.

The regulatory provisions for noise administered by the Councils includes guidelines and protocols issued by the Victorian State government’s EPA underpinned by The Environment Protection Act 2017 [74]. The Act provides an explanation of core concepts, compliance codes, the principles and governance structure that apply to the EPA, as well as the general environmental duty to act and minimise risks of harm to human health, and the environment from pollution or waste. The Act does not reference pedestrians or provide recommendations for data collection methods, mapping, tolerances, or thresholds. The policy used for data collection methods and reporting is the Noise Limit and Assessment Protocol—1826.4 [75]. Its primary function is to determine objective noise limits for new and existing commercial and entertainment venues. The Protocol’s focus is to protect residential amenities from adjacent commercial and industrial land uses and does not provide guidance on acceptable ambient or environmental noise levels impacting pedestrians. They do, however, provide temporal data collection methods and reference values for monitoring noise from a traffic source, assuming the receiver is a resident. The protocols also refer to the ‘Agent of Change’ principle set out in clause 53.06 of the Victorian planning Provisions [76], which assigns responsibility for noise attenuation to new use or development. The principle was created to protect new residents from impacts caused by existing land uses, specifically live music venues, putting some accountability on the receiver of noise to mitigate potential noise complaints. This approach is directly focused on residential amenities offering noise tolerances and limits for habitable rooms, regardless of the external levels of noise with the onus-to-attenuate placed on the designer of new buildings without regard to the interface treatment, which could also be designed to absorb or reflect noise away from pedestrian spaces as well [77,78,79,80]. The State government’s EPA also issued Victoria’s Environmental Protection Regulations that addresses noise emissions from residential premises, commercial, industrial, and trade premises, and music noise from indoor entertainment venues and outdoor entertainment venues and events in Part 5.3, and emissions from motor vehicles in Part 5.6 [81]. This document does not provide guidance for monitoring ambient levels, environmental noise, pedestrians or traffic noise, nor does it recommend the use of noise mapping to inform the noise pollution regulatory framework. The Environmental Protection Regulations [82] and the EPA’s Noise Control Guidelines—1254.2 [83] are primarily used by council officers to resolve noise nuisance using objective metrics that measure emissions of noise without monitoring their impact. Neither policy suggests using soundscape assessments, psychoacoustics, or metropolitan noise mapping.

The State government of Victoria manages most of Melbourne’s transport network, excluding ports and aircraft, which are federally managed, while Local governments are generally responsible for the management of collector roads and local streets. The State is also responsible for hospitals, environmental protection, and planning decisions. However, despite most of the traffic noise emissions being generated along the State-managed roads by the Department of Transport and planning, VicRoads, or privately owned Transurban and Connect East toll roads, addressing nuisances that create health issues falls under Local government authorities’ purview, as stated in the Public Health and Wellbeing Act 2008. The Act mandates Councils to remedy nuisances that create health issues within their municipal districts [84] (p. 88), even though they have only a referral authority to control emissions and do not provide guidelines for measuring noise impacts on a population or recommend metropolitan-scale noise-mapping as a tool to monitor emissions.

#### 3.3.3. Noise and the Environment

The Federal government’s Department of Infrastructure and Transport released ‘Our Cities, Our Future–A national urban policy for a productive, sustainable and liveable future’ which has direct references to environmental noise [85]. It suggests noise is a leading factor that detracts from amenities. It also highlights that low-quality environments correlate with poor health outcomes, including mental health, obesity, and diabetes, with high social and economic costs, citing data from the Australian Bureau of Statistics, which outlines the health and transport implications that emerge from poor amenities [86]. The document also sets out clear objectives on liveability in Chapter 5, which highlights the need to encourage more cycling and walking in poly-centric city structures (such as 20-minute neighbourhoods) by creating safe, well-connected cycling and pedestrian networks with the aim to improve public health, equity, and liveability [85] (pp. 55–63).

While communities are expected to tolerate urban noise to a point, policies are designed to manage, rather than eliminate it [22], with numerous strategies and guidelines designed to protect residential amenities from internal and external noise. These provisions are output-focused, with design and decision guidance aimed at improving development outcomes using objective metrics to monitor and composite materials to attenuate noise from adjacent land uses and traffic. The provisions are prescribed in Volume 2 of the National Construction Code [87] and the Victorian planning Provisions [88] and regulations [89], as per the State government’s planning and Environment Act 1987 [90]. Their focus provides internal noise attenuation guidelines for apartments but offer no guidelines for managing or measuring external noise, which, if available, could leverage acoustic-treatment responses to external areas of new development proposals as a way of demonstrating a response to context or community benefit [91,92] (p. 95).

The National Environment Protection Measure [93] developed under the National Environment Protection Council Act 1994 [94] aims to ensure consistent pollution protection across Australia. However, Perna et al. compare 37 noise policies from Australia, Europe, and North America and claim that Australia’s highest noise annoyance comes from road traffic, but the Australian government protocols have minimal impact on traffic noise [95] (p. 4). In Australia, noise controls are decentralised, with the State governments issuing their own noise collection standards and tolerances that are largely focused on individual vehicle emissions rather than traffic noise as a collective output. Perna et al. attribute this to a lack of accessible noise data collection requirements, compared to other regions, and variations in approaches and noise control tolerances set by different State governments.

Today, the delegation of responsibility for the governance of noise currently sits with Victoria’s Environment Protection Agency [EPA], appointed to oversee pollution issues in Victoria. However, there are numerous other authorities that contribute to governance that need to address noise management and abatement depending on the source. Victoria’s EPA suggests the following actions for people with noise complaints wishing to report noise [96]:Report road traffic noise to the Department of Transport, local councils, or road management companies.Report noise from routine road repair and maintenance to VicRoads, local councils, or road management companies.Report cars with noisy exhausts to the police.Report unreasonable noise from vehicles on private property to local councils.Report noise from public transport services to the respective companies.Report noisy exhausts from large trucks and buses to the National Heavy Vehicle Regulator.Report noise from train and tram maintenance to the EPA’s 24-hour pollution hotline.Noise from major road projects is managed by Major Road Projects Victoria.Report noise from major infrastructure projects to Victoria’s Big Build.Report noise complaints related to licensed premises to the Victorian Commission for Gambling and Liquor Regulation.Report noise from entertainment venues and events to the EPA’s 24-hour pollution hotline.Report concerns about wind energy facility noise to the wind energy facility operator.

The noise-reporting protocols focus on emissions rather than impacts. They do not provide specific guidance for pedestrian complaints per se, but they refer noise complaints from large trucks, public transport, major roads, noisy exhausts, and traffic to an array of authorities positioning the complaints handling protocols far from a pedestrian approach.

The Environment Protection Act 2017 [74] controls unreasonable noise. Under the Act, Victoria’s Environment Reference Standards [ERS] [97] offer guidelines rather than compliance limits. They are outcome-focused and do not provide specific noise limits for enforcement; although, they do apply objective tolerances to land-use classifications, with the Capital City Zone assigned the highest objective noise limits, which exceed the WHO tolerance threshold [98], with the exception of natural environments where the ERS seeks qualitative indicators rather than SPL [97] (p. 14). Part 7.6 of the Act—Control of unreasonable and aggravated noise [74] (p. 196)—provides the controls, conditions, and enforcement regulations. The Act supports the protection of human health and the environment from pollution and waste by identifying values and specifying indicators and objectives aligned with the National Environment Protection Measures [NEPMs] [93]. In the definitions, ‘Pollution’ includes the concept of noise as well as the definition of ‘unreasonable noise’, which requires understanding of the characteristics. Furthermore, Victoria’s EPA recently released the General Environment Duty [GED] in Part 3.2 of the EPA Act 2017 so that any source of noise can be considered unreasonable. This is defined in Section 3 (p. 34) of the Act as having regard to its volume, intensity, and duration; its character; the time and place; how often it is emitted; and other prescribed factors. Assuming the noise is not aggravated, pathways for defining ‘unreasonable’ noise could be approached based on these prescribed factors under Environmental Protection Regulations 2021 (Regulation 120, Regulation 125, Regulation 130, and Regulation 188) [83], or under Section 166 of the Act [99].

Within all the environmental policies reviewed. They all acknowledge the need to protect public spaces from environmental noise, and some focus explicitly on the diminished quality pedestrians experience from environmental noise. None of the documents reviewed recommend metropolitan noise mapping; although, the EPA do recommend using subjective soundscape measurements to assess noise in open public spaces.

#### 3.3.4. Noise and Health

In 2018, the Environmental Health Standing Committee [enHealth] published ‘The Health Effects of Environmental Noise,’ [100], a report aimed at the Commonwealth of Australia with four key recommendations to consider when re-drafting policies related to noise:Recognize environmental noise as a health risk;Promote noise reduction measures;Address noise in planning and development; andFoster research to support policy.

The enHealth recommendations [100] (p. 64) reiterate the WHO recommendations and emphasise that noise should be acknowledged as a critical environmental health issue in Australia while seeking legislative consistency across the three tiers of government and ministerial portfolios. The report found sufficient evidence of a causal relationship between environmental noise, sleep disturbance, and cardiovascular disease to warrant health-based limits for residential areas, though it did not address pedestrian impacts specifically, or the preventative health benefits of active lifestyles. The enHealth document also highlights the need to consider vulnerable groups in noise regulation, including older adults, children, and those with cognitive impairments. It does not refer to other populations such as homeless people, pedestrians, and cyclists that are directly exposed to traffic noise without the benefit of attenuation.

Advocating for improved pedestrian amenities in Australia is The Heart Foundation, a health advocacy group that recognises the health implications of the urban form and the environment. In 2019, they published ‘The Blueprint for an Active Australia’ [101] (p. 17–35). The blueprint recommends the application of sustainable urban mobility by improving safety and security. They advocate for a reduction in both air and noise pollution with the aim to enhance the quality of life and the environment for its citizens, while pointing to the case for behaviour and structural change supporting a reduction in private vehicle dependence, noise and air pollution, energy use, and urban sprawl. They claim that ‘Walking and cycling for recreation and transport, and greater use of public transport, is good for health, the environment and the economy’ [101] (p. 35).

The health policies reviewed and critiqued (including the State government of Victoria’s Public Health and Wellbeing Act 2008 [73] reviewed in Section 3.3.2) highlight the need to protect pedestrian amenities from noise and advocate for using walking as a tool to improve community’s health outcomes. As well as the Public Health and Wellbeing Act 2008, which places Local governments as the responsible authority for addressing noise annoyance and nuisance, enHealth explicitly endorse the WHO recommendations. This provides implied rationale to use metropolitan noise mapping as a research tool to promote noise reduction measures and use subjective soundscape measurements to assess noise impacts in public open spaces.

#### 3.3.5. Noise and Transportation

The National Road Vehicle Standards Act 2018 [102] allows the Minister for Transport to set national vehicle standards to control emissions, including noise. However, the Australian Design Rule 83/00 [103], which outlines the noise limits and measurement protocols for individual vehicles, has no guidance or control over the collective impact of traffic noise, the built form, attenuation structures, or recommendations addressing amenity issue in areas adjacent to carriageways.

The National Austroads research report, AP-R638-20, identifies the National Performance Indicators [NPIs] that have been in place for over 30 years [104]. Reducing noise impacts are one of the environmental objectives used to inform the triple-bottom-line assessment, listing traffic noise exposure as an environmental NPI. While the document highlights that Canada, Japan, and the UK all consider noise exposure as a sustainability and environmental indicator, at the time of publication, no Australian State, including Victoria, has provided performance reporting for the Traffic Noise Exposure NPI. The report provides commentary on compliance with existing policies suggesting that overall, environmental sustainability responses are weak, stating that the Traffic Noise Exposure NPI is typically only evaluated during appraisal, or for project-specific purposes [104] (p.166), rather than for acquiring comprehensive data or performance indicators for precincts, municipalities, or a region. The document refers to pedestrians as a ‘critical component of our transport system and the safety and security of pedestrians is an important focus for Austroads member agencies’ (p. 28). They also released specific guidance on pedestrian planning and design [105] and provided noise wall and barrier design ideas that aim to protect both pedestrians and residential amenities in the interests of health and safety.

In Victoria, VicRoads Traffic Noise Reduction Policy [106] and Traffic Noise Measurement Requirements For Acoustic Consultants [107] provide guidelines that are designed to monitor noise-impacting land uses rather than public space. Victoria’s approach and tolerances require audits be undertaken ‘one metre from the centre of the most exposed window of a habitable room on the lowest habitable level of the building under consideration’ [107] (p. 3) using objective metrics rather than monitoring pedestrian annoyance or impacts in areas where the greatest exposure to noise is likely to occur. These are site-specific protocols designed for acoustic consultants that may be required to perform location audit. They do not advocate for precinct or metropolitan-scale noise maps, even though they could save time and money and be used to inform decision-makers and planners with accessible data rather than funding audits to understand the impact of noise at a specific site over a specific timeframe.

The State government’s Transport Integration Act 2010 [70] emphasises the need to consider equity, user perspectives, decision-making criteria, and triple-bottom-line assessment to ensure that land uses and transport modes are integrated rather than competing. The Act aims to provide an integrated and sustainable transport system in Victoria but lacks references to traffic noise or even the role of pedestrians in the modal hierarchy.

Another policy framework is ‘Movement and Place in Victoria,’ evolving from Smart Roads for effective integrated transport planning [108]. The framework outlines the strategic value of balancing network-wide and localised considerations, allocating modal priority into street classes. Its three planning principles are as follows:People first: Putting transport users at the centre of everything.Outcomes-focused: Delivering more choice, connections, and confidence in travel.One system: Thinking as one system, not individual projects, or modes.

Based on these principles, the Movement and Place framework aims to realise the State’s vision for an integrated and sustainable transport system and inform decision-making for project identification and priority. At the project level, it focuses on user requirements within a street, providing design guidance and impact evaluation. Modules 2 and 4, ‘Network Performance’ and ‘Options Assessment’, suggest noise as a relevant indicator for environmental performance and delineates the principal pedestrian network, mapping out priority areas for projects and protecting amenities [108]. VicRoads is broadly responsible for most of Melbourne’s publicly owned major roads, while local governments are the responsible authority for most of Melbourne’s pedestrian spaces adjacent to the carriageway. Through the Municipal Association of Victoria [MAV], Councils have raised concerns about State and National freight networks and its noise impact on the community. In their Active Transport Policy Statement and Transport Advocacy Strategy they state the following:

‘Councils welcome the opportunity to assist delivery of the Principal Freight Network and ensure local communities are not burdened by trucks, noise, pollution, and inappropriate road network access’.[109] (p. 3)

MAV representatives sit on various committees, including the Road Freight Advisory Group and Essential Services Commission. They advocate for Victoria’s local government authorities and help them to improve community outcomes. Another advocacy group is Victoria Walks. They describe themselves as an evidence-based health promotion charity that is committed to designing safer streets and vibrant places [110]. Victoria Walks seek project funding and provide advice to governments when prioritising investments, developing policies, or building infrastructure designed to encourage walking culture. They see infrastructure as a tool for improved health, commerce, safety, tourism, and transport outcomes. Aligned with the Movement and Place Framework, Victoria Walks also identify that more pedestrian activity will result in reduced noise pollution. They also provide a strategy with pedestrian enabling actions that seek to:

‘…deliver forums and events for decision-makers that profile best and emerging practice and encourage the prioritisation of walking in metropolitan and regional infrastructure and planning’.[111] (p. 3)

The State Safe Pedestrian Program [112] addresses safety from traffic without highlighting the impacts of noise, and the Victorian Pedestrian Access Strategy [113] reveals that 50% of journeys in Melbourne between 400 m and 1 km, and 70% of trips between 1 km and 2 km, are by car. For active people, these short journeys could be more efficiently completed on foot. The strategy’s objectives include improving walking provision, creating pedestrian-friendly environments, increasing walking safety, and integrating walking with public transport.

The transport policies that were reviewed and critiqued present decision-makers with strategic justification and an explicit obligation to protect pedestrians from harm. They identify the significance of walking as a sustainable mode of transport, but do not highlight the need to monitor pedestrian impacts from noise or provide metropolitan noise maps as a performance indicator.

## 4. Discussion

For this policy review and critique, the population is ‘pedestrians’; however, the method could be applied using other vulnerable groups that require tailored infrastructure with the aim to minimise barriers. For other parts of the world, extracting eligible policies can be replicated using the PRISMA–S method and PICo framework to identify relevant policies, research, and advocacy documents in a different context. Not all regions have three tiers of government, and, as such, the mixed methods used in this review may not require using spatial criteria to identify or screen documents. Public sector decision-makers and officers may also find this approach useful for reviewing or evaluating the effectiveness of their own policies by focusing on the implications of a given topic in a multidisciplinary governance structure. This hybrid critique and policy review approach is utilitarian in nature and considers the multidimensional implications of decision-making without the need to understand the physics of sound. For decision-makers that are well versed in their community’s needs, this could be a valuable way of understanding how impacts and subjectivity can be effectively considered.

Over the last century, Melbourne has prioritised vehicular movement, which has eroded pedestrian amenities, exacerbating noise pollution and its health impacts and demonstrating that noise pollution can lead to various adverse health outcomes, including cardiovascular issues, sleep disturbances, and mental health problems [20,37,114,115,116,117]. Despite the health implications of traffic noise, the health benefits of walking, one hundred years of acoustical research, and numerous policies, reports, and standards created by numerous authorities over three tiers of government and global organisations, the problem continues to grow with a growing population and urban densities. With predicted increased freight traffic and population growth, we can expect more government intervention and the need for research and data that highlight the significance of the issue and how noise measurement, in particular, qualitative measures, can be used as a key performance indicator for future structure planning and design using contemporary methods and approaches that measure impacts or outcomes, rather than outputs.

Victoria Walks’ ideas and approaches are aligned with the walking strategies and goals for State and Local governments and authorities. Although not adopted strategies, their advocacy role provides more strategic justification for improving pedestrian amenities, which is considered in community engagement and supported by numerous local governments and corporate supporters [110]. Melbourne’s governance framework does not yet provide explicit mechanisms to protect pedestrian spaces from traffic noise at a Local government level. However, it does provide implicit guidance that advocates for improving pedestrian amenities. The lack of reference to noise in the 23 Council plans reviewed could mean the issue is not considered in decision-making and, therefore, is not seen as a measure of ‘high quality’ and ‘community benefit’, which planning applications could respond to [91]. If environmental noise were to be addressed, applicants could demonstrate how their proposals respond to traffic noise and pedestrian comfort. Existing policies could be amended to highlight the issue and to inform designers and decision-makers where the problem exists for pedestrians, places they could improve, places that need protection, or what a preferred urban character should sound like when designing future aesthetics. This can be accomplished using empirical analysis to create metropolitan noise maps so that designers, developers, and policy-makers can set expectations and support urban structures that protect pedestrian areas from the adverse effects of noise.

At the federal level, the Environmental Health Standing Committee [enHealth] recognises that environmental noise is a health risk while promoting noise reduction measures and fostering research to support policymaking. They acknowledge the need to support vulnerable groups, especially those that are often exposed to higher levels of traffic noise without adequate protection. The report suggests that A-weighted objective metrics be used as a common noise descriptor [100] (p. 4). However, this is not justified as a universal approach. It does not consider inequities for different populations, those that are overly sensitive to noise, a person’s age, nor does it capture perceptions of annoyance for philosophical discussions of noise impact or nuisance [118]. The enHealth recommendations suggest setting national noise goals and mapping in line with the European Union Environmental Noise Directive 2002 [119]. The directive provides a basis for developing measures to reduce noise with requirements to publish strategic noise maps as a reference for the assessment of pollution mitigation efforts, while highlighting Australia’s lack of baseline measures or leadership on the issue. Implementation of national noise goals in Australia could include mapping and monitoring of noise annoyance to promote noise reduction, identify quiet places for the community, areas to avoid, or areas in need of attention. Such an initiative would be an example of research that supports policy, that can be used to highlight environmental noise as a health risk, and as a qualitative reference tool for outcome-focused policies related to transport, the environment, planning, and development. It is worth considering that disadvantaged groups may also be more reliant on active transport for access to social and health infrastructure and could benefit from targeted initiatives.

The multidisciplinarity of this study demonstrates the possibility of displacing responsibility for resolving noise issues. From a user perspective, the EPA complaints handling protocols are difficult to understand, with no single authority governing a response. For traffic noise, reporting could involve multiple responsible authorities appointed within the same road reserve. The pedestrian areas may be the responsibility of the Local government, while the carriageway could be the responsibility of the Department of Transport or even the tramways. This sees the delegation of responsibility vary based on context and whether they are responsible for the source of noise or the receiver of noise, but all the noise protocols are focused on the outputs from the source without regard for subjectivity or understanding impacts other than using the generic decibel scale. The EPA noise reference standards do provide adjustment factors for calibrating objective metrics that consider the variable contexts and urban form. However, outcome-focused policies are more concerned with the impacts experienced by the receiver than the emissions from the source, meaning qualitative and subjective measures are more informative than objective measures of sound. The EPA’s Noise Control Guidelines [83] outline response protocols for noise complaints. However, these are guidelines, not laws, and are used to create conditions on Nuisance Abatement Notices. Currently, the end user of this system would likely struggle to know if the noise levels are within limits or if it is worthwhile intervening. If the end user is a pedestrian in motion, and the traffic is too, the likelihood of capturing the noise offence is minimal, as environmental conditions will change, as will the line-source and -receiver relationship. Universally, the user may not be in a position to describe the noise or explain why it is a nuisance, and it could be argued at arbitration that the noise data protocols effectively capture sound-pressure levels and do not measure noise that by definition contains the subjective measure of annoyance [120,121] or the impact on the complainant at all.

The Movement and Place Framework, while not prescriptive, is useful for collaboration and integrating different modes and provides a collaborative environment where both State and Local governments can work to improve community outcomes. However, multiple modes and authorities bring multiple agendas, raising conflicts that must balance community needs while maintaining the transport network’s integrity. For most of Melbourne’s Council areas, Local governments have both a responsible and referral authority role, and collaboration and discussion could be empowered by the use of metropolitan-scale noise maps that illustrate noise impacts for a priority population.

### 4.1. Identified Gaps

Despite numerous policies aiming to enhance urban walkability and reduce traffic noise from the State and Federal governments, there remains a gap between the State government agenda for improved environmental noise outcomes and strategic directions that enable them at the Local government level. The recently adopted EPA protocols highlight the ‘agent of change’ principle [83] (p. 24), requiring planning applicants to demonstrate noise attenuation designs that protect internal areas while overlooking the need to mitigate noise in public spaces. The framework [82] establishes responsibility for new developments to address external noise to protect habitable spaces from noise generated from adjacent land uses but have missed an opportunity to incentivise external treatments that would attenuate, absorb, or reflect noise away from pedestrian areas.

Another identified gap is the lack of affordable and accessible noise data that illustrate where the problem exists and where its impact needs to be addressed. The European Union’s centralised approach to noise mapping and data collection ensures consistency across member states. Melbourne has the same needs but does not have open-source noise data that can be used to benchmark noise impacts on pedestrians.

Based on this review, no explicit policy mechanism that requires a development proposal to attenuate noise at the interface was found. This gap needs to be addressed so that developers and urban designers can demonstrate ‘community benefit’ or ‘design excellence’ for addressing noise attenuation that protects pedestrian amenities. Developing strategic directions will require further research and comprehensive understanding of acoustics and urban forms to identify appropriate methods and strategic directions that achieve this aim.

### 4.2. Limitations

This paper is limited to the needs of the subject matter and to identifying the decision-making frameworks, noise management, and data collection methods outlined in the aim. Additionally, this study does not provide a comprehensive understanding of acoustics, data collection methods, pedestrian experiences, or explore the intent, extent, or effectiveness of the policies and approaches beyond the strategies and guidelines related to noise or pedestrian amenities within the urban areas of Melbourne, Australia.

The use of the PRISMA–S method over the standard PRISMA method allowed for spatial criteria and an enquiry-driven approach. This method has limitations for replication and evaluation; although, it does facilitate mixed methods such as the content analysis method for evaluation, which was useful in the absence of explicit references to noise in the Council plan’s vision statements and strategic directions. The 23 Council plans identified in the study were reviewed using a relational qualitative content analysis approach. However, only the vision statements and strategic directions were assessed for implicit or explicit references related to noise, health transportation, the environment, and pedestrian amenities. This was due to the variations in policy content where some Council plans included health and well-being plans, while others presented other strategies adopted by the local government area.

The findings of this paper may not apply to other regions with different governance frameworks or urban form; although, the hybrid critique and policy review method may be useful for future comparative policy framework studies or discussion about decision-making and multidisciplinary topics [122].

### 4.3. Further Research

When mandatory requirements, or noise limits on outputs are applied, design and development become more focused on compliance while foregoing the opportunity to innovate, reach community benefit standards, or design excellence. With available technology and data, metropolitan-scale noise maps could be used as a qualitative performance indicator that informs decision-makers on impacts and positive outcomes. This study shows there are substantial data collection methods and noise management protocols that are established and function as intended. However, for the purposes of measuring pedestrian amenities and impacts, more research is required to develop an alternative method for assessing noise impacts as well as measuring negative outputs in Melbourne.

## 5. Conclusions

The aim of this paper is to evaluate the existing governance of noise by critiquing the effectiveness of existing policies and identifying possible insufficiencies. This research concludes that although there are numerous policies from three tiers of government that place pedestrian health and well-being as priority, they lack guidance on how authorities should monitor or respond to environmental noise impacting pedestrian amenities. Federal and State government policies explicitly justify the need to address traffic noise problems. However, Local government Council plans only imply that they need more walkable, healthier environments that facilitate active transport and 20-minute neighbourhoods, without explicit directions on managing noise or its impact in public spaces.

Establishing how effective Melbourne’s governance framework is at protecting pedestrians from traffic-induced noise pollution, the answer varies for State and Local governments. Of all the policies reviewed, three explicitly address the need to protect pedestrian amenities that assign Local governments the responsibility to address noise impacting pedestrians; although, they fail to explicitly reiterate the State government’s strategic directions within Local government decision-making frameworks.

The conclusions demonstrate that there is strategic justification for managing traffic-induced noise pollution to protect pedestrian areas. The 53 documents reviewed and critiqued show a lack of direction or understanding of the benefits of metropolitan-scale noise maps that can inform public sector decision-makers of existing conditions or designers planning for future growth and 20-minute neighbourhoods. Access to new data could raise the issue with professionals from numerous sectors and inform the community, industry, and policy-makers of where interventions would be supported and realise the community’s vision for healthier, more liveable urban spaces, while activating walkable neighbourhoods in Melbourne Australia.

## Figures and Tables

**Figure 1 ijerph-21-01080-f001:**
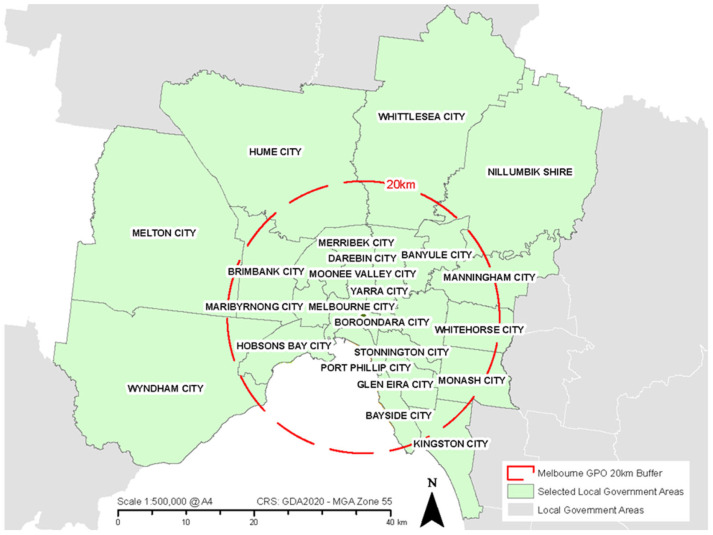
Local government areas included in content analysis.

**Figure 2 ijerph-21-01080-f002:**
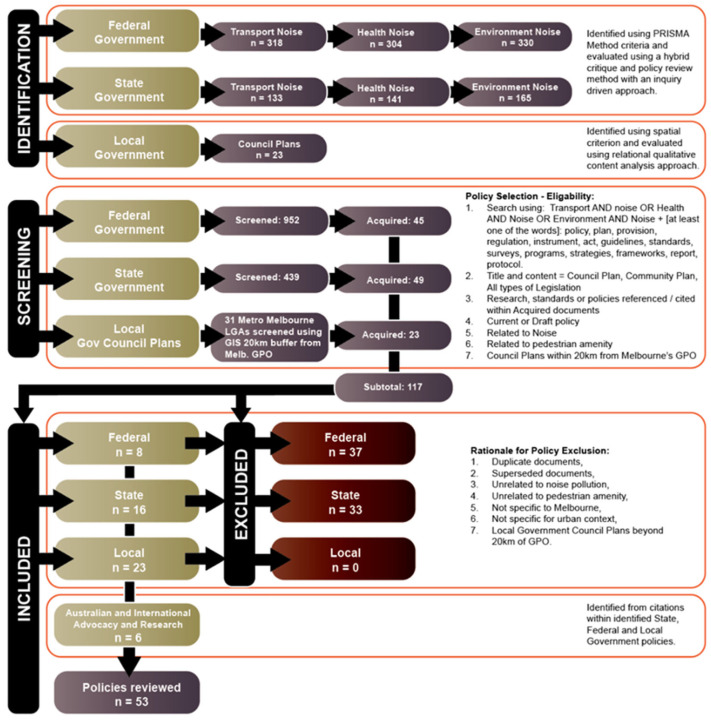
Document-selection flow diagram outlining identification methods, screening for eligibility, inclusion criteria for eligible policies, and exclusion criteria.

**Table 1 ijerph-21-01080-t001:** A sample of the ‘Evaluation of the eligible policies’ document including the evaluation themes and classifications. For the full list, see ‘Appendix A. Each theme was graded with one of three classifications: Green = is present in the document. Amber = is only referred to in a general way. Red = is not present in document.

Focus on Noise [Outputs]	[Explicit] Advocates reduction in Traffic Noise	[Implicit] Advocates improved amenities	Focus on pedestrian amenities	Focus on residential amenities	Focus on community [Outcomes]	Focus on Transport/traffic	Focus on Health	Focus on Environment






**Table 2 ijerph-21-01080-t002:** Extract of the Relational Qualitative Content Analysis classifications from the City of Port Phillip’s Council plan strategies and the results from all 23 Local government Council plans. Strategies that made inferences to the theme or for which noise could be used to indicate the performance of the strategy were highlighted with corresponding colours to the themes of Environment = Green, Health = Blue and Transport = Yellow. If a reference to the theme was identified in the strategic directions, the theme was awarded one point. Totals of each theme were combined to present the likelihood that noise could be used to inform decision-making or the performance of strategies.

Strategy				Environment		Health	Transport
Port Phillip is a place where people of all ages, backgrounds and abilities can access services and facilities that enhance health and wellbeing through universal and targeted programs that address inequities.						1	
A City that is a great place to live, where our community has access to high quality public spaces, development and growth are well-managed, and it is safer and easy to connect and travel within.				1		1	1
Port Phillip is safer with liveable streets and public spaces for people of all ages and abilities to enjoy.				1		1	1
The City is well connected and easy to move around with options for sustainable and accessible transport.				1			1
A City that has a sustainable future, where our environmentally aware and active community benefits from living in a bayside city that is greener, cooler, cleaner and climate resilient.				1		1	
Port Phillip has cleaner streets, parks, foreshore areas and waterways where biodiversity flourishes.				1			
The City is actively mitigating and adapting to climate change and invests in designing, constructing and managing our public spaces to optimise water sustainably and reduce flooding (blue/green infrastructure).				1			
**Content analysis of strategic directions from the 23 Local Government Council Plans**							
**LGA**	**Environment**	**Health**	**Transport**		**Total**		
Banyule	15	11	9		35		
Bayside	9	7	5		21		
Boroondara	15	8	6		29		
Brimbank	11	11	5		27		
Darebin	10	4	4		18		
Glen Eira	12	5	3		20		
Hobsons Bay	7	4	5		16		
Hume City	4	2	2		8		
Kingston	12	11	7		30		
Manningham	30	16	17		63		
Maribyrnong	12	10	9		31		
Melbourne	8	5	4		17		
Melton	5	6	5		16		
Merribek	15	16	14		45		
Monash	4	2	2		8		
Mooney Valley	7	6	5		18		
Nillumbik	13	10	6		29		
Port Phillip	6	5	4		15		
Stonnington	9	11	9		29		
Whitehorse	10	11	6		27		
Whittlesea	9	6	8		23		
Wyndham	8	7	7		22		
Yarra	10	8	10		28		

**Table 3 ijerph-21-01080-t003:** Summary table of the document evaluation.

Topic	Is Present in the Document	Is Referred to in a General Way	Is Not Present in the Document
Focus on noise	34%	9%	57%
Advocates for reduction in Traffic Noise	36%	4%	60%
Advocates for improved amenities	74%	19%	8%
Focus on pedestrian amenities	53%	21%	26%
Focus on residential amenities	81%	15%	4%
Focus on community outcomes	100%	0%	0%
Focus on transport/Traffic	83%	15%	2%
Focus on health	87%	13%	0%
Focus on environment	90%	10%	0%

## Data Availability

All data created for this report can be found in the Appendix A.

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
