# Peer review of "The Governance of Traffic Noise Impacting Pedestrian Amenities in Melbourne Australia: A Critical Policy Review"

_ijerph, 2024, doi:10.3390/ijerph21081080_

Round 1
Reviewer 1 Report
Comments and Suggestions for Authors
The sound environment for pedestrians, which is the theme of this paper, is an important aspect of quality of life in an urban environment. However, I think the structure of the paper is not clear and I did not fully understand the author's ideas.
I would recommend that you organize the structure of your paper and other information before submitting it again. I hope the following comments will be helpful.
1. Comments on the whole paper
1-1. The organization of chapters and sections is complicated to understand.
- Chapters and sections need to be numbered and organized.
- The section title should be more detailed. For example, the following need to be corrected.
i) The section “Research Aim” is on both lines 281 and 604 and does not work well as a section title.
ii) "Identified Gap" (line 679), but it is not clear from the title what the gap is.
1-2. It is difficult to understand the relationship between Results and Discussion (it is unclear what the difference is); it is necessary to clarify what to focus on in the discussion among the Results
1-3. The Conclusion section is rather long, so some of the content should be moved to the Discussion section.
1-4. This paper references various guidelines and documents, but there are so many that it is difficult to understand, especially if the readers are from other countries. Therefore, it would be desirable to have a list of the documents referred to in this paper. The list should also indicate whether the document was issued by an international, national, or municipal organization, and for what purpose.
1-5. Validity of the research question; I think that research questions that can be answered with Yes/No are not always appropriate for this type of review papers. The question posed in this issue, "Is there strategic justification to reduce traffic-induced noise pollution with the aim of to improve walk quality in urban areas of Melbourne?" would need a further approach.
For example, it would be more appropriate to set the following aims;
- Organize and evaluate the existing noise guidelines from the viewpoint of walking amenities, and identify possible insufficiencies
- Discuss what factors and elements should be considered in evaluating the environment for pedestrians
2. Other comments
2-1. Ensure that the terminology used is correct.
i) The following notation should be corrected.
environment noise -> environmental noise
transport noise -> transportation noise
ii) “Use decibels [dB] and Sound Pressure Level [SPL]" in line 203, but it is strange to parallel them since SPL is the name of the evaluation quantity and decibel is the unit of evaluation quantity.
iii) In line 220, there is a description "time-domain analysis [Hz, Time, dB]," but Time is not a unit, so different things are mixed together.
iv) L236 In line 236, the term "subjective measurement" is used, but such an expression is not often seen in noise and environmental assessments. “Subjective evaluation,” for example, seems to be a more appropriate expression.
2-2. In line 274, "Psychoacoustics is the other subjective approach that ~", but I couldn't figure out what psychoacoustics was being contrasted with.
2-3. In line 285, you mention "quality of urban walkability." This is an important concept in the paper, so please explain it clearly, mentioning previous studies, etc.
Author Response
I gratefully respond to the feedback provided and wish to thank you for your time and considered suggestions. The response to this review report are written in red in the attached word document.
As a result, there were extensive changes made and I believe it has been improved as a result.
Kind regards,
David O'Reilly

Reviewer 2 Report
Comments and Suggestions for Authors
The paper wants to demontrate the need of managing traffic noise to improve pedestrian area. Besides the answer is obvious it would be interesting to follow the authors idea to determine which public body can act on that. however the paper structure is completly confusing. Introduction should include aim of the work, there is no need of a separate paragraph. No conclusion should be included in introduction . In introcuction it is not clear if reported conclusions are for the present research or of a cited one. Methods are too short. There are reference to PRISMA and PICo approach which seems to be familiar for the authors but are completely unknown for the most of the acousticians possibly interested to the paper. The session results starts before the reader can understand what's going on in the paper and it is really too long. Table and label should stay in the same page. Discussion paragraph is not expected to have subparagraphs.
In the way the paper is structured cannot catch the attention of people leaving far from the context in which the work is developed. I believe that there is the possibility of making the work more readable and interesting but as is it is not at international level of interest.
Author Response

(The authors gave the same response as above.)

Round 2
Reviewer 1 Report
Comments and Suggestions for Authors
Thank you for your constructive incorporation of my comments.
I think the present manuscript is well revised and the supplementary material is substantial and informative to the reader.